# Biomarker Genes Discovery of Alzheimer’s Disease by Multi-Omics-Based Gene Regulatory Network Construction of Microglia

**DOI:** 10.3390/brainsci12091196

**Published:** 2022-09-05

**Authors:** Wenliang Gao, Wei Kong, Shuaiqun Wang, Gen Wen, Yaling Yu

**Affiliations:** 1College of Information Engineering, Shanghai Maritime University, 1550 Haigang Ave., Shanghai 201306, China; 2Department of Orthopedic Surgery, Shanghai Jiao Tong University Affiliated Sixth People’s Hospital, Shanghai 200233, China; 3Institute of Microsurgery on Extremities, Shanghai Jiao Tong University Affiliated Sixth People’s Hospital, Shanghai 200233, China

**Keywords:** SCENIC, multi-omics, gene regulatory network, microglia, prognosis, immunotherapy

## Abstract

Microglia, the major immune cells in the brain, mediate neuroinflammation, increased oxidative stress, and impaired neurotransmission in Alzheimer’s disease (AD), in which most AD risk genes are highly expressed. In microglia, due to the limitations of current single-omics data analysis, risk genes, the regulatory mechanisms, the mechanisms of action of immune responses and the exploration of drug targets for AD immunotherapy are still unclear. Therefore, we proposed a method to integrate multi-omics data based on the construction of gene regulatory networks (GRN), by combining weighted gene co-expression network analysis (WGCNA) with single-cell regulatory network inference and clustering (SCENIC). This enables snRNA-seq data and bulkRNA-seq data to obtain data on the deeper intermolecular regulatory relationships, related genes, and the molecular mechanisms of immune-cell action. In our approach, not only were central transcription factors (TF) *STAT3*, *CEBPB*, *SPI1,* and regulatory mechanisms identified more accurately than with single-omics but also immunotherapy targeting central TFs to drugs was found to be significantly different between patients. Thus, in addition to providing new insights into the potential regulatory mechanisms and pathogenic genes of AD microglia, this approach can assist clinicians in making the most rational treatment plans for patients with different risks; it also has significant implications for identifying AD immunotherapy targets and targeting microglia-associated immune drugs.

## 1. Introduction

Alzheimer’s disease (AD) is a fatal neurodegenerative disease [1] with a typical pathology of amyloid deposits and neuronal fiber tangles in the brain [2]. It is generally accompanied by a relatively high cellular heterogeneity in microglia [3]. Microglia are the most predominant pathologically disease-rich cells in AD and play a crucial role in AD. However, to date, the internal regulatory mechanisms and causative genes have not been fully identified [4]. Therefore, the identification of transcriptional regulators and the key genes associated with microglia in AD is crucial for identifying AD causative genes and therapeutic targets [5].

In the last decade, the development of RNA-seq data has contributed greatly to the understanding of AD; however, given the random variation of gene expression in single cells, missing data, and technical differences, sometimes the signal-to-noise ratio of snRNA-seq data is even lower than that of bulkRNA-seq data. In addition, it is also difficult to distinguish between direct and indirect regulations, based on the snRNA-seq data only. Therefore, to overcome these problems and to improve the expression of gene regulatory network (GRN) inference, the integration of other data is considered to be a modified approach [5,6]. Compared to single data, multi-omics data hold the promise of capturing the characteristics of AD patients more robustly from multiple, uncorrelated perspectives. A common strategy for identifying multi-omics is to examine each histological stratum individually and then look for overlapping genetic markers between them. Whereas most studies are usually based on differential analysis, studies finding overlapping differentially expressed genes (DEGs) but looking for key regulators with GRNs are rare, and these approaches can be very uncritical. Therefore, future developments should integrate multiple data to construct GRNs. The integration of multi-omics data can reduce the effect of noise and improve performance through the cross-validation of regulation in GRNs with multiple datasets [7,8]. In addition, there are co-expression relationships of genes and regulatory relationships of transcription factors (TFs) with target genes. In most of the published studies, co-expression and GRN were usually independent of each other at two levels; weighted gene co-expression network analysis (WGCNA) mainly discusses co-expression modules and focuses on the modules themselves but fails to analyze the interactions between genes within the modules in detail, which may lose a large amount of regulatory information within the modules.

There are many tools used to build gene networks, including PIDC, ACTION, and SINCERA. SCENIC tops the list in terms of accuracy and biological significance and is currently the tool of choice for building GRNs. Current approaches to integrate multi-omics in AD include the intra-brain combined with peripheral blood, ATAC-seq combined with snRNA-seq data, finding differential genes, and mapping them to another kind of data. However, these integrated multi-omics are more likely to incorporate gene expression than gene activity, much less GRN.

In this paper, we not only propose a method to combine bulkRNA-seq data with snRNA-seq data but also construct GRN modules using single-cell regulatory network inference and clustering (SCENIC) on the basis of both data, filtering the central regulators enriched by both data, which can effectively combine both histological data and improve the utilization of sample information. In the bulkRNA-seq data, we combine WGCNA with SCENIC to construct GRN modules after co-expression network analysis. GRN mapping is depicted for co-expression modules to explore the AD regulatory mechanisms and pathogenic genes within the modules. This can greatly integrate the advantages of both, and it makes up for the shortcoming that WGCNA can only explore inter-module interactions and the sample size of SCENIC cannot be too large. The central regulators of both data were found separately by using PPI; the central regulators of both data were assumed to intersect and were identified, such as *STAT3*, *CEBPB,* and *SPI1*, which were significant in the regulatory mechanism of microglia and have been confirmed to play critical roles in the pathogenesis of AD. Moreover, by modeling the prognoses of these central TFs and assessing the sensitivity to drug treatment of AD patients with different risks, the experiments showed that the sensitivity of some immune drugs was significantly different for high- and low-risk patients. In this study, more accurate potential key genes and their regulatory mechanisms in microglia were identified by constructing GRN through multi-omics analysis of bulkRNA-seq data and snRNA-seq data, which are helpful for understanding more systematically the regulatory relationships of microglia in AD.

## 2. Materials and Methods

In this study, the pre-processing of multi-omics datasets was the first stage, the bilateral Wilcoxon rank sum test was used for snRNA-seq data to screen out AD-associated microglia subpopulations and established GRN to identify regulators by SCENIC; for the bulkRNA-seq data, WGCNA combined with SCENIC were applied to identify the AD microglia-associated modules and establish GRN to determine the regulators. Secondly, those regulators that were identified by both data used PPI to determine the key genes; key genes were selected by taking their intersection. Finally, key genes were used to construct prognostic models and assess drug sensitivity. The experimental flow of this study is shown in Figure 1.

### 2.1. Construction of GRN of snRNA-seq Data Using SCENIC

The snRNA-seq data are widely used in GRN to describe regulatory relationships in cells. There are many tools available, such as PIDC, ACTION, SINCERA, etc., but SCENIC is considered to be superior to other algorithms in the face of massive snRNA-seq data processing. Moreover, GENIE3, as the most reproducible algorithm, has been used in SCENIC, and the RcisTarget database is used in SCENIC to remove indirect targets lacking enriched motifs [9]. Therefore, SCENIC is believed to be an effective method, in our study, by which to construct GRN [9,10].

Firstly, co-expression modules between TFs and target genes were constructed using GENIE3. Secondly, for each co-expression module, the information is imported into the RcisTarget database to identify potential targets that are significantly enriched in transcription factor binding motifs. Finally, the regulator activity in the cells is scored using AUcell, thereby generating a binary activity matrix.

Each p-feature selection task, which is the way in which the GENIE3 method views network inference problems, aims to retrieve the regulators for a specific gene. The method adopts the premise that each gene’s expression levels rely on those of other genes under the same circumstances and was first designed to work with steady-state data [11]. The formula is:(1)$$x_{j}\left(e_{k}\right)=f_{j}\left(x_{-j}\left(e_{k}\right)\right)+\varepsilon_{k},\;\forall j,\;k$$
where $x_{-j}$ denotes the vector with all genes’ expression levels, except for gene *j*, and $\varepsilon_{k}$ represents random noise. GENIE3 assumes that the function *f_j_* only makes use of the gene *j*’s direct regulators expression in $x_{-j}$, i.e., genes in the targeted network that have a direct connection to gene *j*. Recovering the regulatory links pointing to gene *j* are thus equal to the gene of the expression of finding the predictable gene, *j* [11].

Here is a demonstration of how the GENIE3 procedure operates.

For *j* = 1 to p:

Generation of input-output pairs of gene *j* for learning samples.
(2)$$LS_{SS}^{j}=\left\{\left(x_{-j}\left(e_{k}\right),x_{j}\left(e_{k}\right)\right),k=1,\ldots ,M\right\}$$

We learn *f**_j_* from $LS_{SS}^{j}$ and calculate all genes’ confidence levels, w*_i_*,*j* (*i* ≠ *j*), i = 1,...,p, using the feature-ranking technique, except for gene *j* [11]. We use w_i_, *j* as the weight *i* → *j* of the supervisory link.

It should be noted that inputting genes as $LS_{SS}^{j}$ can only be confined to these candidate regulators when a collection of candidate regulators is provided (e.g., known TFs). 

In this paper, the GRNs of microglia and a subpopulation of microglia were constructed using SCENIC to find the key regulators within each GRN network.

The current SCENIC was constructed with default parameters, using the Rcistarget database (based on the hg19–500 bp-upstream-7species.mc9nr and hg19-tss-centerd-10 kb-7species.mc9nr database, with 10 kb around TSS and 500 bp upstream) [12] with constructed regulators for relevant TFs, after removing indirect targets lacking enriched motifs. Target genes significantly enriched for TF binding motifs were retained for further study. Enrichment scores (ES) > 3 and highConfAnnot = TRUE were considered significant regulators for the GRN constructs [12].

### 2.2. Construction of the GRN of bulkRNA-seq Data by Combining WGCNA with SCENIC

For bulkRNA-seq data, the construction of gene co-expression networks enables us to understand the similarity of expression between genes in the disease process and, thus, to obtain information about abnormal cellular roles, functions, and their impact on the pathogenesis of the disease. One of the most widely used methods for gene co-expression network construction is WGCNA [13], which is a correlation coefficient-based network analysis method that converges genes with comparable patterns of expression into the same module. However, WGCNA mainly focuses on the function of the module and fails to analyze the interaction relationship between genes within the module in detail. Therefore, SCENIC is introduced on the basis of WGCNA to construct the GRNs of the same co-expressed genes within the module, to enable the study of the genes within the module more thoroughly. Establishing the GRN of co-expressed genes may be unexpectedly rewarding [13].

BulkRNA-seq data was analyzed by WGCNA modularity, then a soft threshold was calculated, and a scaling-free topology model was constructed. The Pearson correlation coefficient (CC) was >0.7 and data were selected with a threshold of 6. After weight-based filtering, the key modules were finally identified [13]. SCENIC analysis was performed on the key module to construct the GRN and identify the key regulators.

### 2.3. Screening for Common Key Genes of bulkRNA-seq and snRNA-seq Data

For the snRNA-seq data, a number of TFs with high activity and specificity were detected in the microglia subpopulations after the GRN was established using SCENIC. To identify the key factors for these specific TFs, PPI networks were used. PPI action networks were generated using the STRING program (https://string-db.org accessed on 1 February 2022) [14]. The determination of this network allows us to clearly know the weight and degree of interaction between genes and to select the 6 top-ranked genes.

Similarly, for bulkRNA-seq data, after extracting the TFs of microglia-related modules using WGCNA and SCENIC, the PPI network was also used to identify the 6 top-ranked key genes. The PPI network determines key genes, based on centrality metrics (degree, betweenness, closeness).

To identify the key genes more accurately, we took the intersection of key genes from both data sets and obtained the final key genes.

### 2.4. Prognostic Model Construction and Validation

After the identification of key TFs, AD risk profiles were developed using the univariate and least absolute shrinkage and selection operator (LASSO) Cox regression models. The set identified as syn 8691134 was used as the training set and syn 4009614 was used as the validation set, to identify the effect of prognostic AD key genes on overall survival (OS). Genes with a value of *p* < 0.05 were then placed in the LASSO Cox regression analysis. Regression analysis was performed with 1000 iterations of the glmnet [15] R package for gene reduction. Each patient had a risk score assigned to them; they were then split into groups with higher and lower risks.

### 2.5. Drug Sensitivity Assessment

Since testing the effects of drugs on different patients can enable physicians to provide an accurate clinical judgment, the prognostic model was used in our study to forecast the half-maximal inhibitory concentration (IC50) of chemotherapeutic medicines in high- and low-risk patients and to determine the sensitivity of various individuals. Here, the pRRophetic package [15] in R was applied, which allows the application of the pRRophetic model to predict the IC50 of medicines.

This was performed to ensure the effectiveness of the prognostic models constructed by these genes and to infer whether these drugs are meaningful for the treatment of patients.

### 2.6. Statistical Analysis and Operating Environment

The running environment for this experiment is R, version 4.1.2 (released 1 November 2021). The specific R package versions are shown in Table 1. For identifying the differences in cell subpopulations, we used a two-sided Wilcoxon rank-sum test, a log-rank test for KM curves, a Wilcoxon trial for drug sensitivity, and Fisher’s exact test for DO enrichment significance.

## 3. Results

### 3.1. Data Sources and Preprocessing

The snRNA-seq dataset used in this study was downloaded from Synapse (https://www.synapse.org/#!Synapse:syn22079621/) (accessed on 1 February 2022). The snRNA-seq data analysis was performed on prefrontal cortex (PFC) nuclei that were isolated from postmortem human tissue derived from AD and healthy individuals (74–90+ years) (*n* = 11, AD; *n* = 7, normal). The dataset was preprocessed to remove those cells that did not fit the analysis, such as dead cells. The data were preprocessed using the Seurat package; we eliminated those cells with over 5000 or under 200 unique feature counts. By keeping cells with under 5% of mitochondrial readings, dead cells were excluded (percentage mt < 5%). Ultimately, 60,137 cells were eligible for the subsequent analysis. To reduce the effect of data imbalance on sequencing depth, the data overall were positively distributed. The data were normalized using the Sctransform() function in the Seurat package, instead of using the traditional NormalizeData(), ScaleData() and FindVariableFeatures() preprocessing functions. In total, 6000 highly variable genes (HVGs) were selected [16].

The batch processing of the normalized data was investigated using PCA and Harmony algorithms to facilitate the clustering analysis. Ultimately, these important steps were implemented through the RunPCA() and RunHarmony() functions in the Seurat and Harmony packages [16], and the first 20 pc were selected as input.

To identify different cell clusters, we identified cell clusters by a clustering algorithm based on shared nearest neighbors (SNN) modular optimization. K nearest neighbors is calculated and SNN graphs are constructed. Then the modularity function is optimized to determine the cell clusters. The resolution was chosen to be 0.5. In total, 60,137 cell nuclei were clustered, based on 6000 HVGs [16].

The Bulk RNA-seq dataset used in this study was downloaded from (https://www.synapse.org/#!Synapse:syn22079621/) (accessed on 1 February 2022). Bulk RNA-seq data were derived from subjects enrolled in the Religious Orders Study (ROS) or the Rush Memory and Aging Project (MAP), while the dorsolateral prefrontal cortex (DLPFC) data, derived from 90 subjects (44 AD, 46 normal), were used for the downstream analysis. To eliminate data noise, in this paper, the COMBAT function inside the SVA [17] package was used to remove batch effects and normalize the data for TPM.

### 3.2. Cell Type Identification of the SnRNA-seq Data

After the SCTranscform normalization of the data, the 60, 137 cells were retained for subsequent analysis. All cells were divided into 8 cell types (astrocytes cells = ASC, exciting neuronal cells = EX, inhibitory neurons = INH, microglia = MG, oligodendrocytes = ODC, oligodendrocyte progenitor cells = OPC, pericytes/endothelial cells = PER/END). There were also 34 cell clusters (EX1-5, INH1-4, ASC1-4, MG1-3, ODC1-13, OPC1-2, PER/END1-3).

Cell types were annotated using the previous marker genes [18] (ASC for *GFAP* and *SLC1A2*; EX for *SYT1*; inhibitory neurons for *GRIP1*; MG for *LRMDA* and *DOCK8*; ODC for *MBP*, *MOBP*, and *ST18*; OPC for *LHFPL3* and *TNR*) (Figure 2a,b).

### 3.3. Cell Subpopulation Annotation

Cell subpopulation annotations for the above results were performed according to the literature [18]. The annotation of cell subpopulations was based on previously identified marker genes (*LAMP5*/*LINC00507* for the EX1 cluster, *RORB* for the EX2 cluster, *RORB*/*THEMIS* for the EX3 cluster, *FEZF2* for the EX4 cluster, *NRGN* for the EX5 cluster, *VIP* for the INH1 cluster, *PVALB* for the INH2 cluster, *SST* for the INH3 cluster, and *LAMP5* for the INH4 cluster). Similarly, we sought to annotate our glial subpopulations, isolating our astrocyte clusters according to GFAP expression (ASC1 cluster for *GFAP* in *WIF1* + *ADAMTS17*, ASC2 cluster for *GFAP* ^high^ + TNC, ASC3 cluster for *GFAP* ^high^ + *CHI3L*, ASC4 cluster for *GFAP* ^low^ + *WIF1* + *ADAMTS17*). Our microglia clusters are annotated based on the expression of homeostatic and activation markers (*SPP1* + *CD163* for the MG1 cluster, *CX3CR1* for the MG2 cluster, and *ETS1* for the MG3 cluster) [18]. A total of three microglia subpopulations, the MG1, MG2, and MG3 clusters were identified.

The extensive oligodendrocyte annotation is in agreement with the literature [18]; this paper does not deal with oligodendrocytes, so it is not presented in detail.

### 3.4. Identifying Disease-Associated Cell Clusters

To pick out certain cell subpopulations associated with AD for specific study, after snRNA-seq data clustering and the identification of cell types, disease cell clusters were screened for subsequent analysis using a proportional analysis with a two-sided Wilcoxon rank-sum test [18].

Examination of the composition of each cluster revealed that several clusters were significantly over- or under-represented in AD compared to controls (Figure 2c). The ASC3 cluster increased significantly with disease proportion (two-sided Wilcoxon rank sum test, FDR = 2.45 × 10^−3^), while the ASC4 cluster significantly decreased (FDR = 3.52 × 10^−3^). In addition, the proportion of the MG1 cluster also increased with disease (FDR = 2.43 × 10^−2^). Although the results for the MG3 cluster were also significant, the number of cells involved in the MG3 cluster was too small, so the MG1 cluster was chosen as the disease cell subpopulation.

In order to test whether the selected MG1 cluster was associated with AD, we performed differential expression analysis on the MG1 cluster, satisfying |log2FC| > 0.25, with a *p*-value of <0.005, and *p*. adjust of <0.05, yielding 178 disease differential genes. Then, the disease ontology (DO) enrichment analysis of the DEGs in the MG1 cluster by Metascape (https://metascape.org) (accessed on 1 February 2022) (Figure 2d) revealed AD. Therefore, the MG1 cluster was more clearly identified as a disease cell cluster that was significantly associated with AD.

### 3.5. Regulatory Extraction of snRNA-seq Data Based on SCENIC

Considering that regulators play a very important role in AD, the GRN of microglia was further constructed using SCENIC. Some of the more active TFs were listed (Appendix A), such as *ETV5*, *XRC44*, *SMARCA4*, *ELF2*, *CEBPB*, *STAT3*, *SREBF2*, etc.

*ETS2*, *NFATC2*, *CUX1*, *TCF4*, *YY1*, *CEBPB*, *STAT3*, *RB1*, *TAF1*, *BACH1*, *SPI1*, *RUNX1*, and *MITF* seem to be mostly activated in the MG1 cluster. *EP300* and *ELK3* are mainly activated in the MG2 cluster. *SF1*, *BCL6*, *CHD2*, and *CLF2* are mainly activated in the MG3 cluster. Several TFs are also more active in all microglia subpopulations, for example, *ETV5*, *ETS2*, etc. (Appendix A). Both being specific and non-specific TFs, these TFs have an important role in maintaining the microglia-specific transcriptional program throughout the study period.

### 3.6. Analysis of Cell Type-Specific Regulators in Microglia Subpopulations

The degree of TFs specificity shows great variation in each subtype of GRN, and TFs play different functions and roles in AD. Probing for specific TFs facilitates the identification of key genes linked to the disease. Therefore, key TFs in each microglia subpopulation were systematically analyzed. For each pair of regulatory relationships, a regulatory specificity score (RSS) was defined, based on Jensen-Shannon scattering. Specific regulators with the highest RSS values were selected and their functional properties were further examined [19]. Through our network analysis, the specific regulators associated with each microglia subpopulation were identified; the activities of these regulators were highly specific to microglia [20].

For disease cell clusters (MG1 cluster), *ETS2*, *NFATC2*, *CUX1*, *TCF4*, *YY1*, *CEBPB*, *STAT3*, *RB1*, *TAF1*, *BACH1*, *SPI1*, *RUNX1,* and *MITF* are significantly upregulated and *NFIC* is significantly downregulated in the MG1 cluster (Figure 3a). Notably, many of the above TFs are well-known microglia master-regulators. Another well-characterized subpopulation is the MG3 cluster; our network analysis identifies *SF1*, *BCL6*, *CHD2*, and *ELF2* as MG3-specific regulators.

Figure 3b,c below shows the ranking of regulators in the MG1 cluster by RSS. *NR3C1*, *KLF8*, *ATF6*, *CEBPB*, and *STAT3* are highly specific for AD. *MLXIPL*, *NFIC*, *FOSP2*, *FOXO1*, and *ARID3A* are highly specific for healthy samples.

From our results, we can see that a good overview of well-characterized cell-type regulators is provided, and for each cell type, a small number of regulators with very high specificity scores were identified. These conserved and specific TFs can play different roles in the various cell subtypes, which may be related to heterogeneity within the microglia subpopulations.

### 3.7. GRN Construction of Microglia Associated from bulkRNA-seq Data Based on WGCNA

After preprocessing the bulkRNA-seq data, we performed WGCNA analysis on the data and divided these genes into multiple modules (Figure 4a). The black module was found to be enriched with microglial marker genes and AD-related signature genes, such as *SPP1*, *CD163*, *CX3CR1*, *ETS1*, *CEBPB*, *STAT3*, *SPI1*, *NFATC2*, *DOCK8*, *RUNX1*, *ETV6*, etc. Surprisingly, these genes were also enriched in the MG1 cluster in snRNA-seq data.

In addition, to extract and verify the modules with specificity in AD/Con, their correlations were calculated using Pearson correlation coefficients. The degree of correlation between all modules, including the black module, and the clinical traits was also explored. The different modules have different expression patterns and correlations with AD or normal samples. Among them, black (Control/AD: −0.3/0.3), orange (Control/AD: −0.17/0.17), and grey (Control/AD: −0.19/0.19) modules have a relatively high correlation. The black module was found to show the most positive correlation with AD and the most negative correlation with the normal sample, as we expected (Figure 4b).

Therefore, we performed DO enrichment analysis on the black module to verify the correlations between it and the disease. The results showed that the black module revealed significant correlations between the Alzheimer’s disease pathway and the tauopathy pathway (Figure 4c). The above findings directly demonstrated the importance of our selection of the black module as a study.

### 3.8. Regulatory Extraction of bulkRNA-seq Data Based on SCENIC

After constructing the co-expression module, the regulatory relationship between the genes within the module is not clear. Therefore, SCENIC analysis was performed on the co-expressed genes in the black module to construct the GRN. Many regulators were found to be enriched, such as *IRF1*, *STAT3*, *SPI1*, *ELK1*, *EP300*, *ETS1,* etc. Many regulators in the black module overlapped with the regulators of the MG1 cluster. The activity heat map of the regulators in the black module is shown in Appendix A.

### 3.9. Screening for the Common Key Genes and Displaying GRN in Bulkrna-seq Data and Snrna-seq Data

To identify more accurate biomarkers or key genes, after identifying the specific regulators of MG1 clusters in snRNA-seq data and the regulators of the black module in the bulkRNA-seq data, the PPI network was constructed for each of them to screen the central genes and take the intersection.

In the MG1 cluster of snRNA-seq data, we created a PPI network graph via STRING (Figure 5b) and used centrality indicators (degree, betweenness, and closeness) (Appendix A) to assess the importance of genes. *CEBPB*, *SPI1*, *STAT3*, *CUX1*, *YY1*, and *RB1* were identified as the central TFs of the pathological microglia cluster (MG1 cluster).

Similarly, in the black module of bulkRNA-seq data, PPI networks were constructed for all regulators (Figure 5a) (Appendix A). The TFs *LEF1*, *CEBPB*, *SPI1*, *SOX2*, *ETV6*, *STAT3*, *ETS1*, *TCF7L2*, *GATA2*, *STAT5A*, *RUNX1,* and *BCL6* were obtained as the central top TFs.

These TFs in the black module intersected with specific TFs in the MG1 cluster; we found that only *STAT3*, *CEBPB*, and *SPI1* were enriched in both, while many of the target genes that they regulated were similar.

Surprisingly, *CEBPB* and *STAT3* were not only specific in the MG1 cluster but are also specific regulators and disease-associated differential genes that are significantly upregulated in AD (*p*-value = 2.25^−10^, avg_log2FC = 0.262998726).

Figure 5b shows the GRN of *STAT3*, *CEBPB*, and *SPI1* in the MG1 cluster, while Figure 5a shows the GRN in the black module, because the GRN is too large and so only three important genes are shown.

In the MG1 cluster, there are many target genes regulated by *STAT3*, among which *PLEKHH1*, *NEAT1*, *STAT3*, *CPEB4 USP15*, and *SLC1A* are DEGs in the MG1 cluster. When *STAT3* is the target gene, it is regulated by several high-confidence TFs (*ELF1*, *ETV6*, *NRF1*, *SREBF2*, and *STAT5B*). All three central TFs regulate *BCL6*, but the role that *BCL6* plays in AD is currently unknown.

*CEBPB* is a core TF, although it regulates few high-confidence target genes; it regulates *SPP1*, which, in turn, is a marker gene of the MG1 cluster. This implies that *CEBPB* affects *SPP1* and will promote or suppress the inflammatory response.

From these two results, we can see that the GRN constructed in the black module and MG1 cluster obtained similar regulatory results (Figure 6). Many TFs in the black module overlapped with the regulators of MG1 cluster, such as *IRF1*, *STAT3*, *SPI1*, *ELK1*, *EP300*, and *ETS1*, etc. The regulated target genes were also related to those in MG1 cluster, such as *CEBPB* and *BCL6*, etc. Given that there were too many regulators, we only selected the top 100 regulators of weights and enrichment scores of the three central TFs for visualization. We likewise found that *BCL6* was regulated by three central TFs at the same time. In addition to the three central TFs, *CEBPB*, *STAT3*, and *SPI1*; we also found that *BCL6* was regulated by the three central TFs, and there was no doubt that *BCL6* plays a crucial role in AD.

In summary, we combined multi-omics data to build the GRN and found high confidence TFs identified in both data simultaneously (*STAT3*, *CEBPB*, *SPI1*). To assess whether the obtained key TFs were reasonable or associated with AD, we reviewed known AD-related genes from the NCBI official website and found that these specific key TFs were significantly associated with AD. Our study successfully demonstrated that the method in this paper can accurately identify AD-related key TFs, and proved the accuracy and plausibility of our experiment.

### 3.10. Prognostic Model Construction and Validation

In order to verify whether the key genes were effective for prognosis, a univariate Cox regression analysis was performed on the three central genes, two OS-related genes with *p* < 0.05 were screened, and the three genes constructed by the risk model were identified. AD patients in each cohort were divided into high- and low-risk groups (Figure 7a,c) (Table 2). There are 81 individuals in the training set and 54 individuals in the testing set. Based on the median risk score, 81 patients were classified as 40 high-risk and 41 low-risk. In total, 54 patients were classified, with 27 classified as high-risk and 27 as low-risk.

Patients in the high-risk group definitely had a poorer OS than those in the low-risk group, according to Kaplan-Meier curve analysis (log-rank test, all *p* < 0.05). The 1-year survival AUC values, 3-year survival AUC values, and 5-year survival AUC values for the training set ROC curves were 0.530, 0.585, and 0.753, respectively (Figure 7b).

The 1-year survival AUC values and the 5-year survival AUC values for the validation set ROC curves, respectively, were 0.664 and 0.663 (Figure 7d).

### 3.11. Sensitivity Response to Drugs in High- and Low-Risk Patients

To validate the sensitivity of the prognostic model to drugs, according to the pRRophetic, we forecasted the IC50 of several common drugs (Bosutinib, Camptothecin, Cisplatin, Cyclopamine, DMOG, Docetaxel, Gefitinib, etc.) in high- and low-risk patients and found that between the high- and low-risk groups of AD patients, the IC50 values were significantly different (Appendix A).

For example, one drug, Gefitinib, had a higher IC50 in high-risk patients (Figure 8, Wilcoxon trial, *p* < 0.05). Gefitinib may be used to treat low-risk individuals more successfully, as can be observed.

## 4. Discussion

Both bulkRNA-seq data and snRNA-seq data have either more or fewer flaws that can affect the results. Integrating multiple sets of data will reduce the noise of the data itself and improve performance, resulting in a more accurate description of the gene regulatory mechanisms behind diseases and biological processes. In many cases, the GWAS-identified AD risk genes are frequently expressed at extremely low levels in the brain samples, which restricts the extent to which we can examine them. The low expression profile may be due to the heterogeneous brain-cell populations sampled in these studies, diluting specific features including the microglia [21]. We can account for the low expression of microglia genes in the brain (compared to other brain cell types) by studying the co-expression networks and the gene regulatory relationships that underlie them, and we can learn more about the function of these genes and their drug sensitivity to ascertain their role in AD pathology [21].

Therefore, based on multi-omics analysis to find the potential GRN behind AD and determine the potential targets related to AD in GRN, drug sensitivity has become our research direction.

A significant contribution of our research is the identification of three central transcription factors, *CEBPB*, *STAT3*, and *SPI1*, which might be a mediator of AD gene regulation alterations and would have an impact on the pathogenesis of AD.

*CEBPB* is significantly upregulated in the MG1 cluster and AD as a specific regulator of the disease cluster of MG1 and AD. It may affect or regulate AD-related genes and has an integral role in AD. *CEBPB* is considered to probably be the most important regulator in the microglia and can significantly affect neurovascular cell growth and development. Low concentrations of *CEBPB* transcriptional activity inhibit amyloid β pathology and restore cognitive function. This means that *CEBPB* triggers amyloid β pathology and cognitive dysfunction. In addition, *CEBPB* controls pro-inflammatory genes in the microglia and is elevated in AD. It drives an effective pro-inflammatory and neurodegeneration-related gene program. In MG1 cluster, *CEBPB* is considered as a TF of the marker gene (*SPP1*) of the MG1 cluster. Therefore, *CEBPB* affects *SPP1* to show very high specificity in the MG1 cluster and AD. *SPP1* is a marker of neuroinflammation, a chemoattractant, and an adhesion protein that is involved in wound healing [22].

These TFs regulate the genes essential for microglia activation; the promoter and enhancer regions of several cytokines and pro-inflammatory genes include binding sites for these TFs. *CEBPB* is directly associated with the progression of human AD, with elevated levels of the protein appearing in the brains of AD patients, compared to healthy controls. *CEBPB* is highly expressed in macrophages, microglia, and also plays a key role in identifying their activation status [23].

*STAT3* is also a regulator of disease cell-cluster specificity, the AD-specific regulator, the DEG of AD, is significantly upregulated in AD and is regulated by *SPI1* in the black module. *STAT3* is associated with neurodegenerative diseases, wherein neuroinflammation increases the production of pro-inflammatory cytokines and the hyperactivation of microglia. The release of cytokines, which is linked to neuroinflammation in AD, such as tumor necrosis factor (TNF)-α and interleukin (IL)-1β, is influenced by *STAT3* phosphorylation [24].

By connecting to the distal portion of the *CEBPB* promoter, *STAT3* functions as a TF for *CEBPB* as well [25,26]. *STAT3* is present in neurons, the vascular endothelium, astrocytes, and microglia. In response to cytokines, intercellular mediators, and growth factors, it is activated by the phosphorylation of Janus kinase (JAK), which has a variety of biological effects, including cell proliferation, differentiation, and death [21,27,28].

*STAT3* is a canonical inducer of gliosis; the deficiency of *STAT3* not only affects the classical hallmarks of AD pathology but also has a strong positive effect on brain network dysfunction. In the black module, *STAT3* shows the highest activity with its target genes and may have a key role in the severity of AD. It has been documented that rats lacking *STAT3* exhibit significant spatial learning and memory preservation at relatively late stages of the disease. Importantly, these positive behaviors suggest that novel drugs identified by targeting *STAT3* or via increased gliosis are candidates for clinical AD trials in the future. *STAT3* has been nominated in the literature as an AD drug target; the inhibitory agents of *STAT3* reduce neuroinflammation, tau phosphorylation, and the endogenous production of Aβ42 [29]. In subsequent studies to develop drug models for central TFs such as *STAT3*, the drug Gefitinib was found to be the drug of choice regarding *STAT3*. It has a significant effect on the inhibition of *STAT3*.

Although *SPI1* is not a DEG of AD, it was established that *SPI1* is a disease-specific TF by establishing the GRN in the MG1 cluster. *SPI1* also shows very high activity in the black module. The AD gene network’s major hub, *PU.1*, a TF encoded by the *SPI1* gene, is strongly linked to AD pathogenesis. Immune cells, macrophages, and microglia, which have a common embryonic ancestor, all produce *SPI1* at high levels [30]. *SPI1* is a key TF that regulates AD-related genes in the inflammatory response of primary human microglia and microglia [31]. Increased *SPI1* levels lead not only to transcription of AD-related inflammatory genes but also to the dysregulation of genes related to immune response and interferon signaling.

Both in the MG1 cluster and the black module, *BCL6* is simultaneously regulated by these three central TFs, and in the literature [32], it was demonstrated that *BCL6* is a key gene associated with AD immunity. Although the specific role played by *BCL6* in AD microglia is not yet clear, there is no doubt that *BCL6* is likely to serve as a new potential target for AD.

In MG1, *NFIC* is significantly downregulated in MG1, and *NFIC* is located on chromosome 19p13.3. *NFIC* is associated with TGF-β, and TGF-β is a key signaling factor in innate immunity that is associated with APOE pathogenesis in AD. *NFIC* does not appear to be required for normal brain development and it is a functionally unknown transcription factor, the exact cause of which requires further study [33,34].

Another contribution of our study is that the prognostic characteristics of these central TFs are not only reliable for predicted prognosis but also enable the estimation of drug-treatment response in AD patients, which may provide important clinical implications for guiding customized anti-microglia therapy in AD patients and may also have significant implications for targeting microglia-related drugs in the future. Prognostic modeling of central TFs to determine drug sensitivity identified differences in a range of drugs, including Gefitinib, Bosutinib, Camptothecin, Cisplatin, Cyclopamine, DMOG, and Docetaxe, in high- and low-risk patients, which are relevant to AD treatment.

*BACE1* is a potential target for the treatment of AD, and Gefitinib is a potential lead compound for *BACE1* [35]. Epidermal growth factor receptor (EGFR) is an important factor mediating Aβ42 toxicity and is the preferred target for the treatment of Aβ-induced memory loss. The inhibition of Aβ42-induced EGFR activation is an effective treatment for Aβ42-induced memory loss, within which Gefitinib, an inhibitor of EGFR, plays an important role [36]. Gefitinib specifically inhibits the tyrosine kinase activity of the EGFR by interfering with the adenosine triphosphate (ATP) binding site.

There is a small limitation in this paper, in that only the snRNA-seq data and bulkRNA-seq data in the brain were combined, while the blood was not considered. Only the GRN in the microglia was explored, while other cell types were little explored. However, more data, such as spatial transcriptome data and brain imaging data, can subsequently be combined to investigate the rest of the glial cells in a deeper way.

In addition, we have another limitation, in that our data are from only the prefrontal cortex; although the prefrontal cortex is very beneficial for studying AD, it shows some lopsidedness and limitations. It would be desirable to study data from multiple brain cortexes, such as the prefrontal and hippocampus, in future studies. Therefore, combining blood samples with samples from multiple brain regions would be a great step forward for future studies.

## 5. Conclusions

Exploring potential targets has been limited by single histology data when judging key genes by cellular expression, which is far from sufficient. In addition, considering that the bulkRNA-seq data is relatively sparse, snRNA-seq data sometimes show less signal-to-noise ratio than bulkRNA-seq data, while the regulatory mechanisms and risk genes in microglia are not yet fully identified. Single-omics data provide limited information to address complex scientific questions, and a single view is insufficient to capture the full picture of cellular heterogeneity. In this paper, we proposed an approach to integrate multi-omics data based on GRN and combined WGCNA with SCENIC. By using a PPI and centrality index approach, *STAT3*, *SPI1*, and *CEBPB* were identified as the central TFs of microglia in the AD brain. Based on the AD datasets and the related literature, the roles of the extracted TFs in AD were obtained and confirmed. These central TFs are diagnostic markers and therapeutic targets in AD. *CEBPB* promotes amyloid beta pathology and cognitive dysfunction and will regulate target genes regarding inflammation. *STAT3* promotes glial cell production and exacerbates the incidence of AD. *SPI1* regulates primary human microglia and microglial cell inflammatory responses in AD. Prognostic models and drug sensitivity experiments were constructed by these three central TFs, and drugs such as Gefitinib were found to have different sensitivities in different patients. Therefore, the proposed method to construct GRN by the bulkRNA-seq data and snRNA-seq data of multi-omics analysis efficiently provide a greater understanding of microglial regulatory relationships and key genes. This will play an important role in the future prediction of targeting gene-formulating drugs and immune drugs, which will assist clinicians to make more rational and accurate treatment plans.

## Figures and Tables

**Figure 1 brainsci-12-01196-f001:**
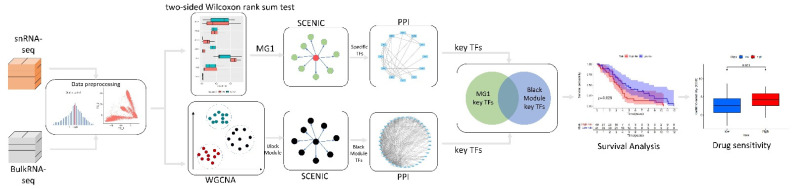
Framework of the experiment.

**Figure 2 brainsci-12-01196-f002:**
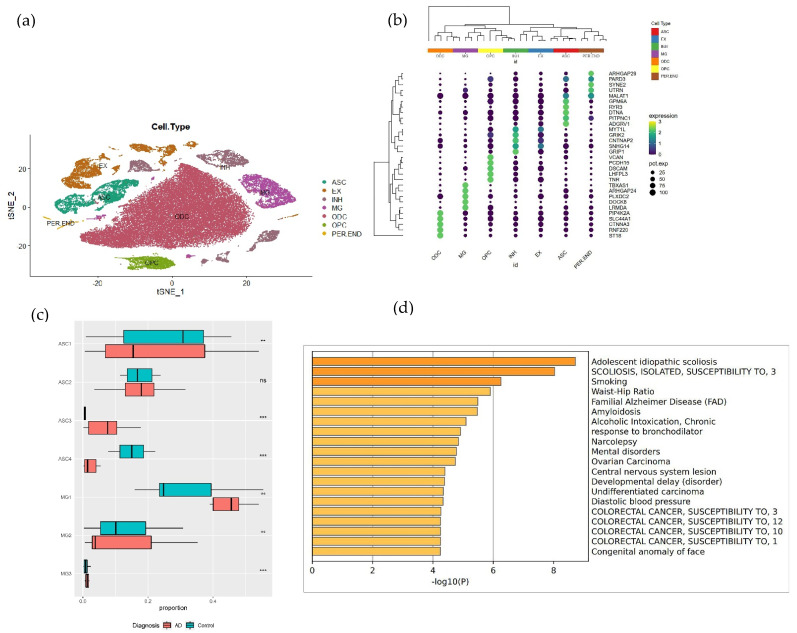
Display of cell types, marker genes, and the extraction of MG1. (**a**) TSNE plot showing 60, 137 cells, mainly for 7 cell types to color. (**b**) Cell type marker gene bubble plot, the size of the bubbles indicates the proportion of genes, from black to yellow indicates low to high gene expression. (**c**) Box plot showed the proportion of each cell cluster and was split by the control and AD samples (“**” represents a *p*-value of <0.05, “***” represents a *p*-value of <0.005, “ns” represents a *p*-value of >0.05). (**d**) DO enrichment analysis of MG1 cluster with −log(*p*) values on the horizontal axis and the pathway list on the vertical axis.

**Figure 3 brainsci-12-01196-f003:**
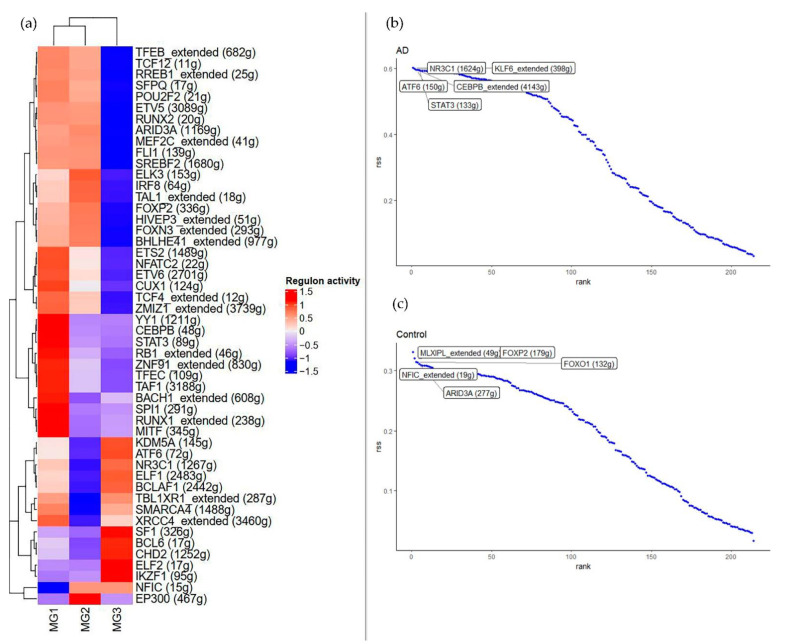
Regulators of differences in MG. (**a**) Heat map of regulator activity, showing the difference in regulator activity between the microglia subpopulations, with negative correlations in blue and positive correlations in red. The values range from −1.5 to 1.5. “Extended” indicates the activity between TFs and all target genes. “No extended” indicates activity between TFs and high-confidence target genes. (**b**,**c**) Ranking of MG1 cluster regulators in AD and normal samples, using the RSS specificity score. “Extended” indicates the activity between TFs and all target genes. (**b**,**c**) are the top 5 specific regulators in AD and Control, respectively.

**Figure 4 brainsci-12-01196-f004:**
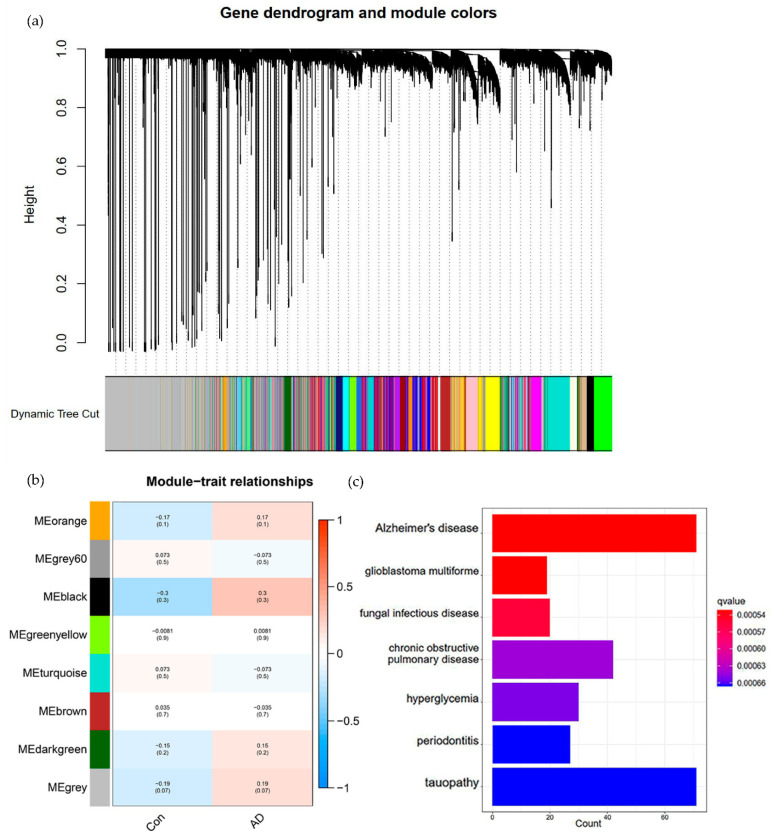
Extraction and verification of the black module, based on the WGNCA results. (**a**) Dendrogram clustered based on a dissimilarity measure (1-TOM). The black module was identified as an AD microglial-related module. (**b**) Module-trait relationships of bulkRNA-seq data. Each row corresponds to a module eigengene, and each column to a trait. Each cell contains the corresponding correlation and *p*-value. Blue to red indicates a negative to positive correlation; the range is from −1 to 1. (**c**) Results of DO enrichment analysis of the black module, with the number of genes in the pathway on the horizontal axis and the list of pathways on the vertical axis, with the q-value (adjusted *p*-value) indicated in blue to red.

**Figure 5 brainsci-12-01196-f005:**
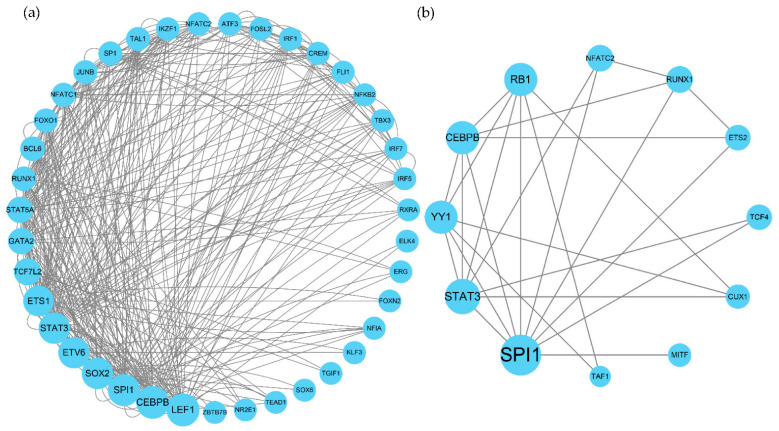
PPI network constructed by STRING and ranked using centrality indicators. The size of the gene depends on the centrality indicators, from small to large indicating increasing centrality. (**a**) Black module regulators building PPI networks. (**b**) MG1 cluster-specific regulators building PPI networks.

**Figure 6 brainsci-12-01196-f006:**
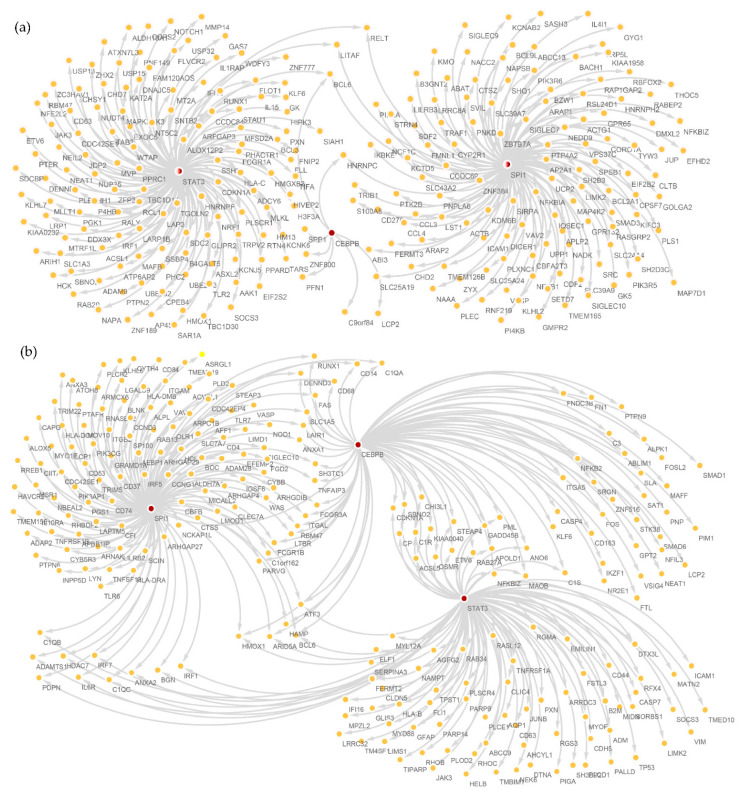
Construction of GRN. (**a**) GRN was constructed by three central TFs of *SPI1*, *CEBPB*, and *STAT3* in an MG1 cluster of snRNA-seq data; the regulated target genes were strictly screened, and all were high-confidence target genes pairs. (**b**) GRN was constructed by three central TFs of *SPI1*, *CEBPB*, and *STAT3* in the black module of bulkRNA-seq data; the regulated target genes were strictly screened, and all were high-confidence target gene pairs.

**Figure 7 brainsci-12-01196-f007:**
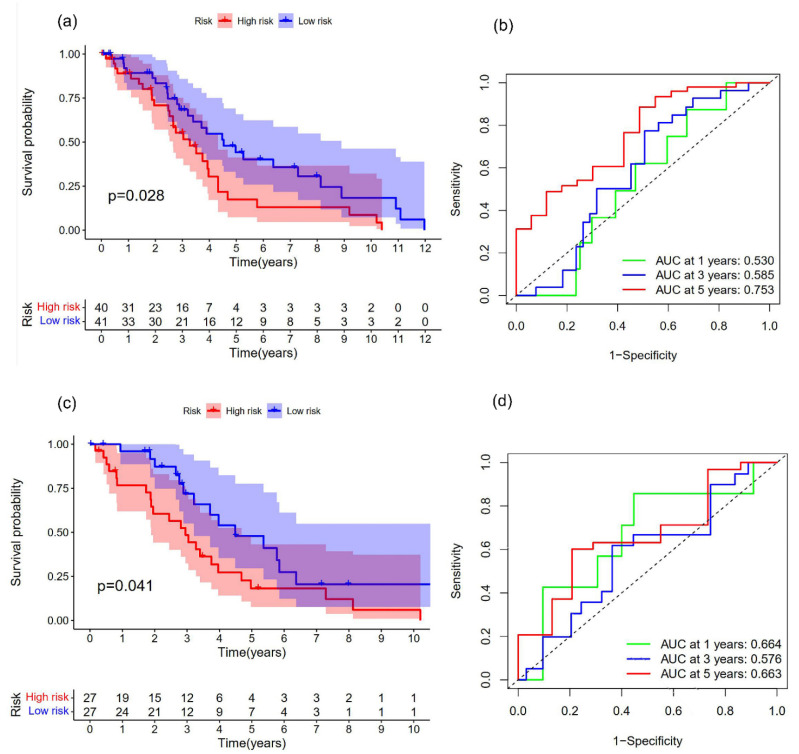
Prognostic model for the training set and testing set. (**a**) The Kaplan-Meier curve is made up through a training set to compare the OS of high-risk and low-risk samples (*p* = 0.028). (**b**) Analysis of the risk score model’s time-dependent ROC curve for forecasting OS of training set at 1, 3, and 5 years. (**c**) The Kaplan-Meier curve is made up through validation sets to compare the OS of high-risk and low-risk samples (*p* = 0.041). (**d**) Analysis of the risk score model’s time-dependent ROC curve for forecasting OS of the testing set at 1, 3, and 5 years.

**Figure 8 brainsci-12-01196-f008:**
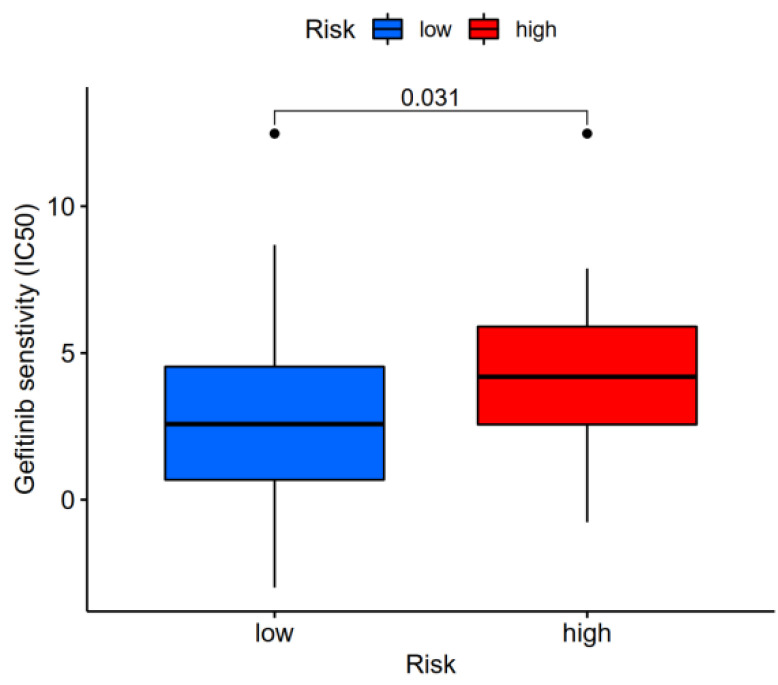
Sensitivity analysis of the Gefitinib drug in high- and low-risk patients, the drug was sensitive to the low-risk group of AD and *p* < 0.05.

**Table 1 brainsci-12-01196-t001:** Version-specific information for R packages.

R Packages	Versions
Seurat	4.0.5
SCENIC	1.2.4
WGCNA	1.70–3
survival	3.2–13
glmnet	4.1–3
pRRophetic	0.5
SVA	3.42.0
tidyverse	1.3.1
patchwork	1.1.1
dplyr	1.0.7
harmony	0.1.0
ROCR	1.0–11

**Table 2 brainsci-12-01196-t002:** Univariate Cox regression analysis of central genes.

Id	HR	HR.95L	HR.95H	*p* Value
CEBPB	0.516515038148349	0.290706574069039	0.917721883269251	0.0242755979166687
STAT3	0.443385886746	0.219963910500021	0.893742269441598	0.0229620332770805
SPI1	0.593750757223216	0.336205951232462	1.04858334723345	0.0724186659708894

## Data Availability

All multi-omics raw and processed data can be found here: https://www.synapse.org/#!Synapse:syn22079621/ (accessed on 1 February 2022). In addition, data can be accessed through our online web application: https://www.ncbi.nlm.nih.gov/geo/query/acc.cgi?acc=GSE174367 (accessed on 1 February 2022). Gene expression profiles and clinical information for the prognostic model construction training set were obtained from the AMP-AD Knowledge Portal database (raw count data are available online at https://www.synapse.org/#!Synapse:syn8691134 (accessed on 1 February 2022), and validation datasets were obtained from the AMP-AD Knowledge Portal database (normalized data are available online at https://www.synapse.org/#!Synapse:syn4009614) (accessed on 1 February 2022).

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
