# Peer review of "Biomarker Genes Discovery of Alzheimer’s Disease by Multi-Omics-Based Gene Regulatory Network Construction of Microglia"

_brainsci, 2022, doi:10.3390/brainsci12091196_

Round 1
Reviewer 1 Report
Authors combined snRNA-seq data and bulkRNA-seq data to construct co-expression gene networks and obtained cell clusters and subpopulations in Alzheimer's disease. The model containing key genes(STAT3,SPI1,SEBPB) was linked with overall survival and drug sensitivity.
The paper is valuable for scientific community.However, some points have to be impoved.
Major points:
Quality biomarkers are characterized with AUC >0.7, thus the survival predict model does not pass validation.
An intermediate data should be provided(e.g., TPM values after COMBAT)
The introduction must cover the current advances in the are investigated (gene networks and models in Alzheimer's disease, multiomics approach in Alzheimer's disease)
Minor points:
Majority of detailes are underdescribed, that makes the manuscript unclear. Also, the text quality should be improved(punctuation, mistypes, etc)
-Each abbreviation has to be deciphered the first time that it is used.
-Each tool/ database/ method has to be cited.
-The link to the datasets are not clickable, it will be better to indicate the resource and the dataset IDs.
-Line 293 - "negative correlation in blue and positive correlation in red" in the figure caption. However, a value varies from -1.5 to 1.5 in the figure legend that does not contain a correlation coefficient but another value. The description should be improved.
-Abbreviations in figures have to be deciphered(e.g., subpopulations names, cell types, etc). What is "_extended" in labels?
-Punctuation and style should be improved in lines 226-229 , 216-218.
-Gene names have to be given in italic.
-What 34 cell clusters were obtained?
-"highConfAnnot = TURE" looks like a mistype.
-A duplicated legend in Fig.2b
-What value was cut off with threshold of 6?
-Figure 7 does not contain any legend or keys, it is unclear what biological conclusion of this clustering.
-Labels in supplementary figures are not readable.
Author Response
Response to Reviewer 1 Comments
Point 1: Quality biomarkers are characterized with AUC >0.7, thus the survival predict model does not pass validation.
Response 1: In many articles, there is no clear range of AUC values, but of course the higher the better. In many articles, values greater than 0.5 or 0.6 are also accepted.
Point 2: An intermediate data should be provided(e.g., TPM values after COMBAT)
Response 2: BulkRNA-seq and snRNA-seq preprocessed data are provided in the supplemental file.
Point 3: The introduction must cover the current advances in the are investigated (gene networks and models in Alzheimer's disease, multiomics approach in Alzheimer's disease)
Response 3: Information on the current progress has been added to the introduction.
Point 4: The link to the datasets are not clickable, it will be better to indicate the resource and the dataset IDs.
Response 4: The data address has been changed correctly.
Point 5 : Line 293 - "negative correlation in blue and positive correlation in red" in the figure caption. However, a value varies from -1.5 to 1.5 in the figure legend that does not contain a correlation coefficient but another value. The description should be improved.
Response 5: All have been modified
Point 6 :-Abbreviations in figures have to be deciphered(e.g., subpopulations names, cell types, etc). -What is "_extended" in labels? -Punctuation and style should be improved in lines 226-229 , 216-218. -Gene names have to be given in italic. -"highConfAnnot = TURE" looks like a mistype. -A duplicated legend in Fig.2b
Response 6: All have been modified
Point 7: What 34 cell clusters were obtained?
Response 7: All have been modified .The text goes on to describe in detail which specific 34 cell clusters are involved, and it is also stated that this thesis focuses only with microglia, for other cell types are not included as part of the discussion.
Point 8:What value was cut off with threshold of 6?
Response 8:A soft threshold of 6 is derived from the WGCNA algorithm, and a co-expression network of genes is better when it is 6, which is just one parameter.
第9点:
图7不包含任何图例或键,不清楚这种聚类的生物学结论是什么?
响应9:图7只是WCCNA的一个中间过程,主要可视化一些共表达模块。在这里,WGCNA被比作集群。这是一个类比,实际上没有聚类的生物学结论
Reviewer 2 Report
The manuscript is titled "Biomarker genes discovery of Alzheimer’s diseaseby Multiomics-based gene regulatory network construction of Microglia".
This article presented a framework to integrate multi-omics datasets to discover biomarkers, broadly translatable to several biological systems. The Introduction states the objectives of the work and provides an adequate background. The Materials and Methods section is clear. The integration section explores the essentiality of multi-omics for the comprehensive understanding of AD. The Result section is well presented. Overall a good manuscript of multi-omics data integration.
However, the authors should address these comments/questions: 1. Add a paragraph about the potential limitations of the study. 2. Did the authors find doublets and removed them in the analysis? 3. Add a paragraph in the method section to include all the statistical tests included in this study. 4. Improve the figures (1, 2, and 5) quality. 5. Some text edits:
line 257, DO should be spelled as disease enrichment at first instance line 328, Bulkrna-Seq and Snrna-Seq should be corrected or follow the same whenever they appear. line 560, the author contribution section should be updated. Thanks.
Author Response
Point 1: Add a paragraph about the potential limitations of the study
Response 1: Limitations have been added at the end of the discussion.
Point 2: Did the authors find doublets and removed them in the analysis?
Response 2:In this paper, only nCount_RNA, nFeature_RNA, and mitochondrial ratio were filtered. Because there is no doublets -related data in the data, doublets are not considered.
Point 3: Add a paragraph in the method section to include all the statistical tests included in this study
Response 3:statistical test has been added in the last part of the method, and the R runtime environment and the version of the R package have been added. All statistical tests in the text are statistical test functions that come with the R package, and many are embedded in the functions.
Point 4:
line 257, DO should be spelled as disease enrichment at first instance line 328, Bulkrna-Seq and Snrna-Seq should be corrected or follow the same whenever they appear. line 560, the author contribution section should be updated
Response 4:All the above points are changed.
Reviewer 3 Report
In the current paper, Gao et al. reanalyze the snRNA-seq and bulk RNA-seq data previously published in S Morabito et al. Nat. Genetics 2021. With a distinct analysis pipeline and different cell types of interest (microglial cell), they found that central transcription factors STAT3, CEBPB, SPI1, and their regulatory network display unique coexpression patterns associated with AD patients group in both snRNA-seq and bulk RNA-seq. And they established an AD prognostic model based on their previous results, of which they validated the sensitivity to drugs. Most of the analyses look comprehensive and convincing. However, the authors did not put their writings, methods, and figures in a mindful framework, significantly reducing the quality of the works. Unfortunately, I cannot recommend publication based on the current version.Major concerns: 1. Methodological details of results should be in the method session. And the results should deal specifically with what the analyses indicate (e.g., Data sources and preprocessing Cell subpopulation annotation, and so on). It is extremely difficult to go through the steps of the analyses with the current flow of writing. Additionally, there are no clear citations along with key packages used in the paper (e.g., SCENIC, SVA, WGCNA, glmnet). The authors actually show the analysis workflow in Figure one clearly. it will significantly improve the readability if they can add their analyses pipeline step-by-step in the methods in parallel with their workflow. Furthermore, it is also beneficial to add your survival analysis and drug sensitivity analysis into the experimental workflow figure.
2. The methods are relatively unclear. It is understandable that it may be hard to provide all the details since this paper involves many methods. However, since this paper is a data reanalysis project, it is required to make your workflow fully reproducible and transparent. - For example, it is unclear how they assign the risk group in survival analysis. Furthermore, the demographics of the current training and validation sets should be provided. It is also unclear how they deal with covariates in Cox regression. The genes involved in the prognostic model are also convoluted. A table is required as all the subsequent analyses were based on it. - As far as I understand, I don't see any description of FDR control for multiple tests in the method description (e.g., univariate Cox regression on genes). This is important in most NGS applications.
4. Furthermore, since this is an intensive bioinformatics workflow, it is important to also upload your annotated processing codes in Github or Zenodo. It will significantly enhance the transparency, reproducibility, and readability of the analyses. A table of packages and their versions used in the workflow should be provided in the supplementary table.
5. Figures of which results are related should be grouped. It is hardly readable with 15 figures scattered through the paper. I would suggest reformatting figures into six figures. Figure 1 deals with experimental workflow and a table of demographics or sample information. Figure 2 deals with single cell analyses pertaining to the justification of focused analyses on MG1 cluster. Figure 3 deals with MG1-related regulators. Figure 4 deals with WGCNA and the black module in bulk RNA-seq. Figure 5 GRN analyses combining both bulk- and sn-RNA-seq. Figure 6 deals with the prognostic model and drug sensitivity.
6. The main issue with the figures is that the authors somehow completely ignored visualization of expression pattern in MG1 cluster shown in snRNA-seq or black module shown in bulk RNA-seq. For example, Figure 3 shows the proportion of the MG1 cluster presented in AD vs. Control. However, it is unclear how the gene expression are different within the cluster. Similarly, Figure 7 also didn't show expression patterns within the Black module. And what immediately they can improve is to color and scale size of the network figure shown in Figures 9-10 based on the differential expression pattern comparing AD vs. Control and the significance of FDR value.
7. The authors show little care for the "Data Availability Statement". It is unacceptable to copy the statements from the source paper and paste almost the same here. Furthermore, it is not acceptable to claim "your web app." The authors must revise it. And the authors didn't properly cite the paper (S Morabito et al. Nat. Genetics 2021) in their method and data availability statement. It is also important to mention the agreement of the data use for this dataset.
Minor concerns: 1. Inconsistent labeling of single-cell RNAseq (scRNA-seq, snRNA-seq). Such mistakes should have been easily identified and corrected by the authors before submission. 2. Inadequate citations (e.g., SCENIC). To facilitate other researchers to reproduce your results, It is extremely important to cite your references in the method session even if you have noted them in the main text before. 3. There are very few comments on the different/similar outcomes of the current analyses compared to S Morabito et al. Nat. Genetics 2021 (such as ext. Fig. 6 specific to dysregulated TFs in MG). 4. In Figure 3, what is the meaning of the asterisks. The statistical test and significance level should be mentioned in the legend. 5. In Figures 11-12, the authors should highlight or create another panel to show the overlap between the two GRN networks. 6. It is completely invisible for supplementary Scheme 2. The author should improve the quality of that figure.
Reference: 1. Morabito, Samuel, Emily Miyoshi, Neethu Michael, Saba Shahin, Alessandra Cadete Martini, Elizabeth Head, Justine Silva, Kelsey Leavy, Mari Perez-Rosendahl, and Vivek Swarup. “Single-Nucleus Chromatin Accessibility and Transcriptomic Characterization of Alzheimer’s Disease.” Nature Genetics 53, no. 8 (August 2021): 1143–55. https://doi.org/10.1038/s41588-021-00894-z.
Author Response
Point 1: there are no clear citations along with key packages used in the paper (e.g., SCENIC, SVA, WGCNA, glmnet). it is also beneficial to add your survival analysis and drug sensitivity analysis into the experimental workflow figure.
Response 1: Partial citation information has been added and survival analysis and drug sensitivity have been added to the flow chart.
Point 2: it is unclear how they assign the risk group in survival analysis. Furthermore, the demographics of the current training and validation sets should be provided. It is also unclear how they deal with covariates in Cox regression. The genes involved in the prognostic model are also convoluted. A table is required as all the subsequent analyses were based on it. - As far as I understand, I don't see any description of FDR control for multiple tests in the method description (e.g., univariate Cox regression on genes). This is important in most NGS applications.
Response 2: The demographic data of the training set, validation set, and assigned risk groups have been illustrated, providing univariate cox regression tables for the prognostic genes.
Point 3: A table of packages and their versions used in the workflow should be provided in the supplementary table.
Response 3: The R runtime environment and the version information of specific R packages are provided (Table 1). The code can only provide the main program, part of the code involves innovation inconvenient to provide.
Point 4: Figures of which results are related should be grouped. It is hardly readable with 15 figures scattered through the paper.
Response 4: In response to the comments you provided, the images in the full text were revised. However, I did not put prognosis and drug sensitivity together at the end, for the reason that it would not be easy to layout the images if they were put together.
Point 5:The main issue with the figures is that the authors somehow completely ignored visualization of expression pattern in MG1 cluster shown in snRNA-seq or black module shown in bulk RNA-seq. For example, Figure 3 shows the proportion of the MG1 cluster presented in AD vs. Control. However, it is unclear how the gene expression are different within the cluster. Similarly, Figure 7 also didn't show expression patterns within the Black module.
Response 5:
The reviewer was trying to show how the extracted MG1 and black modules differ from the other modules at the gene expression level. It certainly is different, and Figure 3 first shows the differences in expression between the different subgroups in AD versus control, and then Figure 5 shows the differences in gene expression between the different isoforms. Looking at it this way, it is true that Figure 7 does not show the expression pattern of the Black module, which was chosen for the reason that this module is enriched in microglia Marker genes, and here only the gene-to-module correlation or expression matrix for all modules, including Black, is provided.
Point 6 :what immediately they can improve is to color and scale size of the network figure shown in Figures 9-10 based on the differential expression pattern comparing AD vs. Control and the significance of FDR value.
Response 6: Figures 9-10 do not directly correlate with the differential expression pattern of AD and control, and I guess you mean the importance of gene centrality indicators, so I changed the image so that it is sorted by size by centrality indicators. And tables of centrality metrics are provided (Supplementary Tables S1,S2).
Point 7: The authors show little care for the "Data Availability Statement". It is unacceptable to copy the statements from the source paper and paste almost the same here. Furthermore, it is not acceptable to claim "your web app."
Response 7: Revised data availability statement and website address.
Minor concerns:
Point 1:Inconsistent labeling of single-cell RNAseq (scRNA-seq, snRNA-seq). Such mistakes should have been easily identified and corrected by the authors before submission.
Response 1:The whole text is modified to snRNA-seq. but there are actually some differences between scRNA-seq and snRNA-seq, so there was a distinction before.
Point 2:There are very few comments on the different/similar outcomes of the current analyses compared to S Morabito et al. Nat. Genetics 2021 (such as ext. Fig. 6 specific to dysregulated TFs in MG).
Response 2:Added information about dysregulated NFIC in MG to the discussion.
Point 3:Inadequate citations (e.g., SCENIC). To facilitate other researchers to reproduce your results, It is extremely important to cite your references in the method session even if you have noted them in the main text before. In Figure 3, what is the meaning of the asterisks. The statistical test and significance level should be mentioned in the legend.
Response 3:Both of the above 2 points are modified.
Point 4:In Figures 11-12, the authors should highlight or create another panel to show the overlap between the two GRN networks.
Response 4:The figure is trying to show the intersection of the two kinds of data, these two graphs are the GRN graph after filtering, which can obviously see the common TFs CEBPB STAT3 SPI1.
Reviewer 4 Report
This paper from Gao et colleagues, used different approach to integrate multi-omics data to provide new insights into the potential regulatory mechanisms and pathogenic genes in AD microglia. The paper is well structured, and the analysis of the data are very thorough. The only concern I have is that they used only data from one brain region. To make the conclusions stronger I would at least analyse data from hippocampi, another region majorly affected in patients with Alzheimer’s disease.
Author Response
point1:The only concern I have is that they used only data from one brain region. To make the conclusions stronger I would at least analyse data from hippocampi, another region majorly affected in patients with Alzheimer’s disease.
Response 1:The reviewer made a very good point and I agree with it. My initial intention was to find brain data from the same person, but so far I have not found BulkRNA-seq data and SCRNA-seq data with both prefrontal cortex and hippocampus. I believe it would be a great enhancement to my article if such data were available.
Round 2
Reviewer 3 Report
The authors address most questions comprehensively. However, for point 5, I would insist that the authors should at least demonstrate that in bulk-RNAseq analysis, the relationship between the black module and the trait (AD vs. Control). It is not surprising that the black module, enriched with microglial marker genes, will share signature genes with the microglial cluster identified by again, microglial marker genes in single-cell RNA-seq. The focus of MG1 is justified by AD-associated proportional difference. However, it is important to also demonstrate that the black module displays a differential expression pattern between AD vs. control to close the loop. It would be at least nice to put the results as a supplementary figure for figure 4 or some panels in figure 4. But overall, the authors did a great job with the revision.
Minor:
1. Please fix the typo from "Black model" to "Black module"
Author Response
Dear Editors and Reviewers:
Thank you for your letter and for the reviewers’ comments concerning our manuscript entitled “Biomarker genes discovery of Alzheimer’s disease by Multi-omics-based gene regulatory network construction of Microglia. Those comments are all valuable and very helpful for revising and improving our paper, as well as the important guiding significance to our researches. We have studied comments carefully and have made correction which we hope meet with approval. Revised portion are marked in red in the paper. The main corrections in the paper and the responds to the reviewer’s comments are as flowing:
Point 1: However, for point 5, I would insist that the authors should at least demonstrate that in bulk-RNAseq analysis, the relationship between the black module and the trait (AD vs. Control). It is not surprising that the black module, enriched with microglial marker genes, will share signature genes with the microglial cluster identified by again, microglial marker genes in single-cell RNA-seq. The focus of MG1 is justified by AD-associated proportional difference. However, it is important to also demonstrate that the black module displays a differential expression pattern between AD vs. control to close the loop.
Minor: Please fix the typo from "Black model" to "Black module"
Response 1: Yes, thanks for the reviewer’s comments, we accept the reviewer’s suggestion. We provided the analysis of module-trait relationships between the black module and the trait (AD vs. Control) with figures in the revised manuscript. The different expression patterns and relationships of the Black module in AD/Control are that the Black module presents a positive correlation in AD and a negative correlation in Control. Thus, it was demonstrated that the black module showed a different expression pattern between AD VS. control to close the loop. (3.7 section, line 324-331, 339, 340-342. Figure 4b)
In addition, "Black Model" has been revised to "Black Module".
